# Predicting Low-Level Childhood Lead Exposure in Metro Atlanta Using Ensemble Machine Learning of High-Resolution Raster Cells

**DOI:** 10.3390/ijerph20054477

**Published:** 2023-03-02

**Authors:** Seth Frndak, Fengxia Yan, Mike Edelson, Lilly Cheng Immergluck, Katarzyna Kordas, Muhammed Y. Idris, Carmen M. Dickinson-Copeland

**Affiliations:** 1Department of Epidemiology and Environmental Health, School of Public Health and Health Professions, University at Buffalo, Buffalo, NY 14260, USA; 2Department of Community Health and Preventive Medicine, Morehouse School of Medicine, Atlanta, GA 30310, USA; 3Geographic Information Systems, InterDev, Roswell, GA 30076, USA; 4Department of Microbiology, Biochemistry, and Immunology, Morehouse School of Medicine, Atlanta, GA 30310, USA; 5Department of Medicine, Morehouse School of Medicine, Atlanta, GA 30310, USA

**Keywords:** lead exposure, machine learning, geographic prediction, primary prevention

## Abstract

Low-level lead exposure in children is a major public health issue. Higher-resolution spatial targeting would significantly improve county and state-wide policies and programs for lead exposure prevention that generally intervene across large geographic areas. We use stack-ensemble machine learning, including an elastic net generalized linear model, gradient-boosted machine, and deep neural network, to predict the number of children with venous blood lead levels (BLLs) ≥2 to <5 µg/dL and ≥5 µg/dL in ~1 km^2^ raster cells in the metro Atlanta region using a sample of 92,792 children ≤5 years old screened between 2010 and 2018. Permutation-based predictor importance and partial dependence plots were used for interpretation. Maps of predicted vs. observed values were generated to compare model performance. According to the EPA Toxic Release Inventory for air-based toxic release facility density, the percentage of the population below the poverty threshold, crime, and road network density was positively associated with the number of children with low-level lead exposure, whereas the percentage of the white population was inversely associated. While predictions generally matched observed values, cells with high counts of lead exposure were underestimated. High-resolution geographic prediction of lead-exposed children using ensemble machine learning is a promising approach to enhance lead prevention efforts.

## 1. Introduction

Significant neurocognitive deficits occur at blood lead levels (BLLs) <5 µg/dL [1,2,3], and there is no known safe threshold for lead exposure [4]. As early as 2006, Gilbert and Weiss argued to lower the actionable threshold to 2 µg/dL [5]. In 2021, the Centers for Disease Control and Prevention (CDC) lowered the threshold from 5 to 3.5 µg/dL (the 97.5%ile of the general pediatric population in the U.S.) [6]. While mean BLLs have dropped significantly among U.S. children, from 13.7 μg/dL in 1976–1980 [7] to 0.80 μg/dL in 2011–2016 [8], the 97.5 percentile BLL has not changed in over a decade [9]. Between 2007 and 2010, over 500,000 U.S. children aged 1–5 years [10] had BLLs ≥5 µg/dL, and an estimated 2 million children ≤5 years had BLLs ≥2.98 µg/dL [11]. As the effects of lead are often irreversible [12], exposure prevention continues to be an important public health goal.

No standard lead screening or testing policies exist across the U.S. [13,14]. In New York State (NYS), an environmental assessment of the home is not required until BLL ≥5 μg/dL [15]. In contrast, Georgia requires a home inspection if a child’s BLL is ≥3.5 µg/dL [16]. Preventive programs such as the Erie County Childhood Lead Poisoning Primary Prevention Program in NYS provide community resources for lead remediation. However, these resources are limited to zip codes with high numbers of children with elevated BLLs (≥5 μg/dL) [17], potentially missing children at lower levels of exposure. Furthermore, allocating limited resources to large geographic areas, such as zip codes, is inefficient. We suggest a predictive approach using a high-resolution raster grid may help allocate limited resources to smaller geographic areas for exposure prevention.

Geographic-based predictive models of lead-exposed children have faced numerous challenges. In 2020 Potash and colleagues created a random forest model to identify locations of Chicago children with BLL of ≥6 µg/dL enrolled in the Women, Infants, and Children (WIC) program. Their model performed well, but they used a high threshold (6 µg/dL) and a sample of children at high risk for lead exposure, thus potentially limiting the generalizability [18]. In 2021, Liu and colleagues used a stacked ensemble to predict BLLs at the individual level using a dataset of children monitored for lead exposure in Broken Hill, Australia. This was another well-performing model, but the dataset included children from 1991–2015. Sources of lead exposure have changed significantly since the 1990s, and updated models are needed to better predict the changing landscape of exposure sources [19]. Lastly, Lobo and colleagues predicted the presence of children with BLLs ≥5 µg/dL within specific zip codes [20]; however, zip code classifications are too large to implement targeted preventative measures with limited resources. In sum, it is necessary to develop updated models using a greater geographic resolution to minimize the time and costs associated with zip code-level intervention.

Previously, we identified individual- and community-level factors that distinguished children with BLLs <2, 2–5, and ≥5 µg/dL [21]. As we previously showed [21], metro Atlanta has a significant number of children exposed to both lower and higher levels of lead within the Healthy Homes for Lead Prevention Program database. We concluded that a greater number of U.S. children aged 2–6 years were at risk for low-level lead exposure than the previously estimated [22] 500,000 children. We now propose an ensemble machine learning approach to predict the number of children with BLLs 2–5 µg/dL and ≥5 µg/dL in a high-resolution geographic raster grid of the metro-Atlanta area to further refine our ability to predict, down to a small geographic area, where at-risk children reside. Our goals were two-fold: (i) to create and evaluate a predictive model of the number of children with BLLs ≥2 to <5 µg/dL and ≥5 µg/dL using 1 × 1 km raster cells in metro-Atlanta, (ii) identify differences in global variable importance when predicting the number of children with BLLs ≥2 to <5 µg/dL and ≥5 µg/dL. Successful prediction in a high-resolution geographic space with different levels of lead exposure will help identify smaller geographic areas for targeted intervention.

## 2. Materials and Methods

### 2.1. Study Sample

The initial sample comprised surveillance data of 491,973 children aged <19 years with geocoded addresses from the Georgia Department of Public Health’s Healthy Homes for Lead Prevention Program (GDPH-HHLP) [22]. Surveillance data were collected from multiple healthcare providers and laboratories throughout the 20-county metro Atlanta region between 2010 and 2018. In the rural U.S., sources and prevalence of lead exposure can differ [23], and children are often under-tested [24,25]. Furthermore, our raster cells would include vastly different population sizes between urban and rural areas. Therefore, we restricted our sample to children from metro Atlanta’s urban and suburban zip codes. We excluded children if they did not have a venous blood draw, were >72 months, did not have a complete residential address, or were duplicated within the same calendar year. Our final sample included 92,792 children.

### 2.2. Study Sample Raster Grid Creation and Outcomes

First, the area, including 20 counties surrounding metro Atlanta (henceforth ‘20-counties’), was converted into a high-resolution raster grid using the 1984 World Geodetic System (WGS84). We wanted our raster cells to represent a large enough sample of children in each cell. No prior literature exists to establish raster cell size for predicting the number of children with lead exposure. Therefore, we used a 0.001 × 0.001-degree resolution (~1 × 1 km cells) a priori. Our primary outcomes were the number of children in each raster cell testing at each threshold (≥2–<5 µg/dL and ≥5 µg/dL). We only included raster cells from which a child was tested for lead. Children were sampled from 6591 1 km^2^ raster cells, each containing our primary outcome and predictor attributes. The number of children sampled from each raster ranged from 1 (true for 1593 cells) to 750 (1 cell) (Q1 = 2, Median = 4, Q3 = 12).

### 2.3. Study Sample Raster Grid Creation and Outcomes

We chose 13 predictors commonly associated with lead exposure in the literature. Because we hope to validate our model in other cities, we chose predictors that could be readily extracted for many U.S. cities. Furthermore, we did not exclusively include causal predictors of lead exposure, as our purpose was primarily predictive, not causal.

#### 2.3.1. Predictors from the U.S. Census

We used 2010 census data because sample collection began in 2010, and census data collection was challenging during the beginning of the 2020 COVID-19 pandemic [26]. Demographic characteristics from census blocks within the 20 counties were assigned to raster cells whose centroid fell within the census block. Rasterization of census area data is common, and applications include predicting the spatial distribution of the Aedes albopictus [27], understanding the population road use [28], and identifying inequitable geographic distributions of toxic exposures [29].

Of the 12 study predictors, six were census-based. Lead exposure is often concentrated in poor areas of high disadvantage, among older housing stock containing lead paint, and in areas with a high percentage of persons of the color [30,31,32,33]. Thus, our chosen predictors were the percentage below poverty, the percentage black, the percentage white, the percentage of houses built before 1980, the percentage of >25-year-olds without a high school diploma, and the Gini index. The Gini index is a measure of census block income inequality. The index ranges from 0 to 1, with 0 indicating high equality and 1 extreme inequality. High inequality occurs when most of the income in a census block is concentrated among a few individuals [34]. The total housing units by census block were also included to account for differences in population density.

#### 2.3.2. Environmental Protection Agency (EPA) Toxics Release Inventory (TRI)

The EPA toxics release inventory (TRI) contains point locations of larger industries that manage and release toxic chemicals into the environment (via air, water, or landfill). There are 770 toxic chemicals in the TRI program, including lead [35]. These point locations were converted into a density raster grid using a kernel density function. A raster cell with a higher number of nearby industry locations received a higher TRI score. We created kernel density plots for the raster grid using point locations of (i) all facilities included in the TRI [Total TRI], (ii) facilities that release toxic chemicals into nearby water [Water TRI], (iii) facilities that release toxic chemicals into the air [Air TRI], and (iv) geographic areas where facilities release toxic chemicals on land [Land TRI]. Separating industries by location of toxic chemical release helps fine-tune potential sources of lead exposure for children with ≥2–<5 µg/dL and ≥5 µg/dL.

#### 2.3.3. Crime Index and Road Network Density

As lead exposure is often found in high-crime areas [36] and our previous publication demonstrated a relationship between crime and lead exposure, we also included our previously constructed 2019 crime index as a predictor [21]. Briefly, this crime density index is compiled by ArcGIS using the FBI Uniform Crime Report at the census block level. Higher crime index values indicate greater crime compared to a national average. Raster cells were assigned the crime index value of the census block they fell within based on the centroid of the raster cell. As lead exposure is common in high-trafficked areas [37,38], we also included road network density as a predictor. A road shapefile was downloaded from the Georgia Department of Transportation (GDOT), and a kernel density function calculated the density of roads in each raster cell. The final dataset contained 6591 rows of data representing a ~1 × 1 km raster cell, each with 13 predictors.

### 2.4. Study Sample

Following Ahmed et al.’s (2021) approach using explainable artificial intelligence (XAI), our predictive model was a stacked ensemble of three base learners: an elastic net generalized linear model (GLM), a gradient-boosted machine (GBM), and a deep neural network (DNN) [39]. The elastic net GLM is a regression classifier that finds the optimal penalties to shrink the beta coefficients of variables not predictive of the outcome. GBM is a type of random forest model that aggregates predictions from generated decision trees. GBM differs from a random forest by giving additional weight to high-performing decision trees. DNN is a neural network that uses forward and backward propagation to create notes using given predictors to optimize performance. These base learners were chosen because they allowed for the specification of a highly skewed count distribution. Ahmed et al. (2021) also demonstrated the predictive power of a stacked ensemble approach compared to geographically weighted techniques in a geographic space, supporting our decision to use a stacked ensemble with XAI. A stacked ensemble (or super learner) uses a meta-learner to combine predictions of base learners into a meta-prediction [40,41]. Model agnostic methods were subsequently applied to explain the relationship between the top four predictors and our outcomes using partial dependence plots (PD). A flow chart of data processing and model training is found in Figure 1.

#### 2.4.1. Training, Validation, and Test Datasets

To prevent overfitting, our sample was randomly split into 50% training, 25% validation, and 25% test datasets [39,40], with 10-fold cross-validation on the training set. While there is no gold standard for data splitting, especially for count data, we chose this percentage partitioning because we wanted many non-zero counts represented in the training and validation sets for hyperparameter tuning. A test dataset was used to evaluate model performance on an unbiased hold-out dataset [41,42,43].

#### 2.4.2. Base Learners and Stacked Ensemble Model

All model training and interpretation methods were implemented using the h2o package in R [44]. The non-normal distribution for counts of BLLs ≥2 to <5 µg/dL and counts of BLLs ≥5 µg/dL warranted the specification of a Tweedie distribution, allowing for inflated zeros and count data. Hyperparameters for each base learner (GLM, GBM, DNN) were tuned with 10-fold cross-validation in the training dataset and tested against the validation dataset. We used a Random Grid Search (RGS) to maximize efficiency to find optimal hyperparameters in all base learners. RGS samples from all combinations of hyperparameters rather than testing all possible hyperparameter combinations. The base learner hyperparameter combination with the lowest root mean squared error (RMSE) in the training dataset was selected. Stopping rules were specified to indicate when the RGS has reached an optimal hyperparameter combination. Following Ahmed et al. (2021) [39], we used 0.001 as our stopping tolerance and 2 as our stopping rounds, as a <0.001 change in RMSE would not be a meaningful model improvement. Using a stopping tolerance also limits the number of generated models at each iteration and prevents overfitting. All final hyperparameters for the base learners are provided in the results. We used a GLM meta-learner for our ensemble with 10-fold cross-validation using the predictions from our best-performing base learners.

#### 2.4.3. Model Performance, Predictor Importance, and Partial Dependence Plots

Both RMSE and mean absolute error (MAE) were reported across all learners to evaluate model performance. Using the test dataset, we present the ensemble permutation-based importance and model agnostic importance and partial dependence (PD) plots for the top 4 predictors. Permutation-based importance represents how important a variable is for prediction by reporting the RMSE loss if the variable were to be removed. PD plots show the global model-predicted number of children with BLLs ≥2 to <5 µg/dL and ≥5 µg/dL across different predictor values controlling for all other predictors in the model. These plots help interpret the relationship between each predictor and outcome. A final model-predicted geographic map of percent BLLs ≥2 to <5 µg/dL and ≥5 µg/dL is also presented alongside maps of observed values. Lastly, we included a simple median of the count data to examine if our model outperforms an educated guess.

## 3. Results

Descriptive statistics for the analytical sample are presented in Table 1 by BLL threshold. There were slightly fewer females (48.4% female) for the full sample, and children were about 2 years of age on average (mean = 25.9 months, SD = 14.8). A large proportion did not disclose race (43.5%) or identified as Black (33.9%). A moderate percentage were from urban zip codes (22.5%), and a large majority received Medicaid (73.4%). Most sampled children had BLLs ≥2–<5 µg/dL (n = 52,522, 56.3%), followed by <2 µg/dL (n = 38,124, 40.9%) and ≥5 µg/dL (n = 2146, 2.3%). Among those with BLLs ≥2–<5 µg/dL and ≥5 µg/dL, a large proportion did not report race (61.2% and 43.2%, respectively) or reported black race (22.9% and 28.1%, respectively).

Descriptive statistics of the raster cells are presented in Table 2. Two observations among children ≥5 µg/dL were >42 standard deviations above the mean (SD = 2.56; count observations = 109,145). Therefore, we truncated these two observations to the next highest count value (count observation = 36). Across the 6591 cells, the number of children with BLLs ≥2–<5 µg/dL and ≥5 µg/dL ranged from 0 to 63 and 0 to 36, respectively. The mean number of children in a raster cell with BLLs ≥2–<5 µg/dL was 7.2 with a median of 2. The mean number of children with BLLs ≥5 µg/dL was 0.29 with a median of 0.

### 3.1. Model Building, Performance, and Interpretation

RMSE and MAE across training, validation, and test datasets of the base learners (predicting BLLs ≥2–<5 µg/dL and BLL ≥5 µg/dL) are presented in Table 3 using the full variable set and top five predictors. Based on the lowest RMSE in the test dataset, the ensemble performed equally or better than all base models for predicting BLLs ≥2–<5 µg/dL and BLL ≥5 µg/dL. The ensemble model outperformed the simple median for both BLLs ≥2–<5 µg/dL and BLL ≥5 µg/dL on RMSE. However, it was close to the median MAE for BLL ≥5 µg/dL.

#### 3.1.1. Predicting BLLs ≥2–<5 µg/dL

The hyperparameters for the base learners in the BLLs ≥2–<5 µg/dL ensemble included: (GBM: 563 trees, column sample rate of 0.95, max depth of 15, a minimum of five rows, and a Tweedie power of 1.4; GLM: alpha of 0.5, lambda of 0.1, and a Tweedie variance power of 1.05; DNN: 3 hidden layers each with 50 units, a “Rectifier” activation function, and a Tweedie variance power of 1.95). Variable importance for the number of children with BLL ≥2–<5 µg/dL is presented in Figure 2. In order of importance, the top four predictors were: EPA Air TRI density, Percent of the population below the poverty threshold, percent White, and the crime index.

Partial dependence plots for the ensemble are presented in Figure 3. This figure also provides a histogram for the predictor and data density shown with vertical tick marks (“data rug”) in the figure background. The most important variable, EPA Air TRI density, had a positive relationship with the number of children with BLLs ≥2–<5 µg/dL. The percent of the population below the poverty threshold was also positively related to the number of children with BLLs ≥2–<5 µg/dL. The percentage of the population that is White was inversely related to the number of children with BLLs ≥2–<5 µg/dL. Crime index appeared to have a threshold effect, with greater numbers of children with BLLs ≥2–<5 µg/dL after an index value of ~300.

To demonstrate our model geographically, we plotted the observed and predicted values of all raster cells in Figure 4. The model-predicted counts largely mapped onto the observed counts, predicting generally high and low numbers of children with BLLs ≥2–<5 µg/dL. However, our predicted counts were generally lower than the observed counts. This was especially true for counts above the >95%ile in the observed data, which were predicted lower than reality (i.e., the maximum predicted number ≥5 µg/dL was 227 vs. the observed maximum of 522).

#### 3.1.2. Predicting BLLs ≥5 µg/dL

The hyperparameters for the base learners in the BLLs ≥ 5 µg/dL ensemble included: (GBM: 567 trees, column sample rate of 0.7, max depth of 11, 1 minimum number of rows, and a Tweedie power of 1.05; GLM: alpha of 0.25, lambda of 0.1, and a Tweedie variance power of 1.05; DNN: 4 hidden layers each with 10 units, a “Rectifier” activation function, and a Tweedie variance power of 1.15). Variable importance for the number of BLLs ≥5 µg/dL is presented in Figure 5. In order of importance, the top four predictors were: EPA Air TRI density, percent White, road network density, and the crime index.

Partial dependence plots for the ensemble are presented in Figure 6. The most important variable, the EPA Air TRI density, had a positive relationship with the number of children with BLLs ≥5 µg/dL. The percentage of the population that is white was slightly negatively related to the number of children with BLLs ≥5 µg/dL. The greatest decline occurred in the <25% white population. Road network density was positively related to the number of children with BLLs ≥5 µg/dL. Lastly, the crime index appeared to have a threshold effect, with greater numbers of children with BLLs ≥5 µg/dL after an index value of ~300. However, this occurred in a sparse area of the observed data. The crime index pattern was similar for BLLs ≥2–<5 µg/dL.

We plotted predicted counts alongside the observed counts for BLLs ≥5 µg/dL in Figure 7. The predicted and observed counts were similar; however, high observed counts (>95%ile) were predicted lower by the model (i.e., a maximum predicted number ≥5 µg/dL was 18 vs. an observed maximum of 36).

## 4. Discussion

As there is no safe threshold of lead exposure, predicting which geographic areas contain children with low BLLs is of public health concern. Our goal was to create a predictive model of the number of children with BLLs ≥2–<5 µg/dL and BLLs ≥5 µg/dL in the metro-Atlanta area with recent BLL data (2010–2018). Using a high-resolution raster grid with counts of exposed children, our predictive model yielded promising predictive potential. Using smaller geographic units to predict where children may be exposed can support primary prevention efforts. To date, this is the first published attempt at the prediction of child lead exposure using a high-resolution raster grid.

### 4.1. Model Performance: Strengths and Limitations

This represents a first attempt at predicting the number of children with BLLs ≥2–<5 µg/dL and BLLs ≥5 µg/dL in a high-resolution raster grid. In general, the model characterized raster cell counts well, as demonstrated in Figure 4 and Figure 7. Downtown and northeast Atlanta, identified by the ensemble learner, have been previously identified as hot spots for lead exposure, including DeKalb, Gwinnett, and Cobb counties [21]. The ensemble learner also predicted more precise geographic locations. In metro Atlanta, Cobb, Fulton, DeKalb, Hall, and Gwinnett counties have been identified as “high risk” for lead poisoning [45]. Our model confirmed these counties as high-risk, finding hotspots in each county. However, our model provides a more precise geographic estimate than a county-wide classification of “high risk.” For example, our model revealed that Gwinnett’s northern and southern areas have higher counts than central Gwinnett. Also, only southern areas of Hall County contained high counts of lead-exposed children. This is promising for future implementation of similar predictive models.

Our ensemble model performed well overall. First, there was no evidence of overfitting or underfitting. The difference between RMSE and MAE in the training, test and validation datasets were similar for the ensemble model. Second, the ensemble model had equal or lower RMSE and MAE values compared to the base learners. Third, our model outperformed an educated guess about the number of children in a raster cell with BLL ≥5 µg/dL and BLLs ≥2–<5 µg/dL. The RMSE demonstrated a much better performance of the ensemble compared to a simple median. However, for BLLs ≥5 µg/dL, the MAE for the simple median was close to the ensemble. In this case, the RMSE is a better indicator of model performance. RMSE penalizes the difference between observed and predicted values more than MAE. Given the skewness of our count data, RMSE is a better indicator. We also note that the simple median is informed by pre-existing knowledge of the distribution of the counts in each raster cell. Our ensemble model only uses environmental predictors to determine the counts in each cell and knows nothing about the location or number of children with BLLs ≥5 µg/dL or BLLs ≥2–<5 µg/dL. A simple median may perform better than random guessing.

It should be noted that high counts were not predicted well. This is especially apparent for BLLs ≥5 µg/dL. While a hot spot of children with BLLs ≥5 µg/dL is detected in northeast Atlanta, the predicted counts underestimate the actual number of children with BLLs ≥5 µg/dL in this area. Different predictors may be needed to improve predictive performance for high-count raster cells. Perhaps alternative sources of exposure are present that are not represented in our predictor set. We might improve future iterations of our model by including alternative geographic predictors such as estimated soil lead, land use change [46], or vacant properties [47].

### 4.2. Important Predictors among BLLs ≥2–<5 µg/dL vs. BLLs ≥5 µg/dL

Three common predictors were identified in both the BLLs ≥2–<5 µg/dL vs. BLLs ≥5 µg/dL models: EPA air TRI, percent of the population that is White, and crime index. Additionally, for BLLs ≥5 µg/dL, road network density was a top predictor, while for BLLs ≥2–<5 µg/dL, percent below poverty was a top predictor. The direction of associations in the partial dependence plots also demonstrates that the model reflects previously published observations of child lead exposure. Interestingly, the EPA air TRI was the top predictor in both models. The EPA air TRI was created using the kernel density of EPA-flagged facilities that release toxic chemicals into the air. An ecologic analysis of U.S. counties (2000–2007) revealed that 1.24% of children had BLLs ≥10 µg/dL in counties with high air lead levels compared to 0.36% in low air lead counties [25]. The high importance of EPA air TRI may be unique to the metro Atlanta area, and further investigation is needed. Crime index and percent of the population below poverty are proxies for socioeconomic conditions that lead to exposure to lead sources, including lead paint, plumbing, contaminated soil, and other spatiotemporal variables for which there is limited data [1,2]. Finally, a study leveraging ensemble machine learning to predict BLLs at the individual level also found racial health disparities with higher levels in Aboriginal than in non-Aboriginal populations [19].

While there are some differences in the rank order of importance for predictors between the BLLs ≥2–<5 µg/dL and BLLs ≥5 µg/dL models, these differences were minimal. Local interpretations of ensembles may reveal differences in predictor importance across observations or geographic areas. For example, heterogeneity in risk factors for lung cancer resulted in differences in predictor importance at the local level [39]. We presented global importance measures, and future investigation of local importance measures may clarify predictor importance across the Atlanta metro area.

### 4.3. Strengths and Limitations

In general, the transportability of our models to other U.S. cities warrants further investigation, particularly in rural areas. This study only used urban and suburban areas in the metro Atlanta region. Therefore, implementing our current model in rural areas would be inappropriate, as rural areas likely contain fewer children than urban and suburban areas in each raster cell. Furthermore, many children in rural areas are missed by current BLL screening. In rural North Carolina, an estimated 30% of children > 30 months with BLLs ≥3 μg/dL were missed by current screening practices [24]. A predictive model of BLL counts specific to rural areas would be of significant public health interest.

Furthermore, our sample contained a large percentage of children on Medicaid (73.4%). The percentage of the population on Medicaid in Fulton County was 16.1%, DeKalb 19.2%, and Clayton 28.4% in 2015 [48]. It is possible that children on Medicaid were more likely to be screened and included in the GDPH-HHLP database. Using an oversampling of children enrolled in Medicaid has both strengths and limitations. First, as children in poverty are more likely to be exposed to lead, using a high Medicaid-enrolled sample might make predictions more sensitive to children in poverty. However, the model’s generalization to high-income areas may result in poorer model performance. We should also note that we did not know the self-reported race for a large percentage of children. While individual-level data were not part of our predictive model, this does make the assessment of generalizability difficult. However, the known racial and socioeconomic distribution suggests that our source data was diverse. Our model represents an important first step for high-resolution lead prediction using a very large sample with specific address information.

## 5. Conclusions

To our knowledge, this is the first attempt to predict the number of children with low-level lead exposure using a high-resolution raster grid. Future development of high-resolution machine learning models can be implemented to prevent lead exposure within a more refined geographic area.

## Figures and Tables

**Figure 1 ijerph-20-04477-f001:**
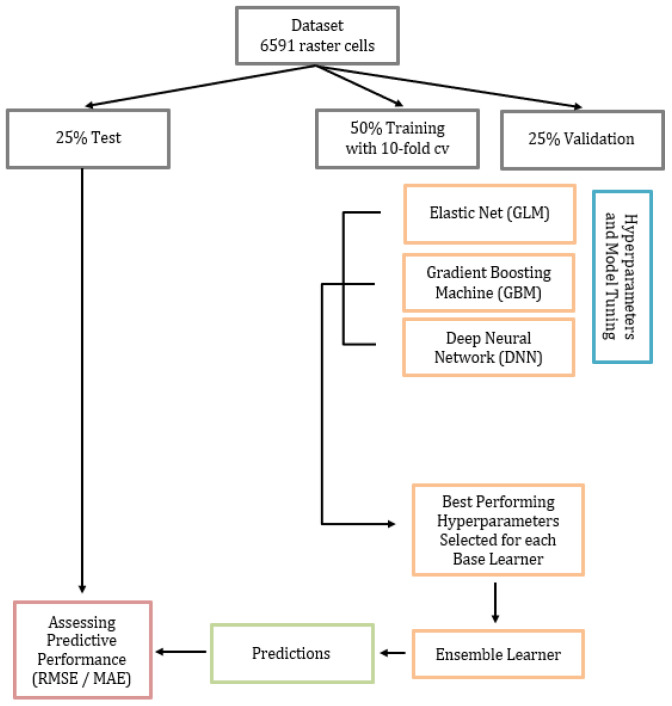
Data processing and model building flow for the ensemble ML.

**Figure 2 ijerph-20-04477-f002:**
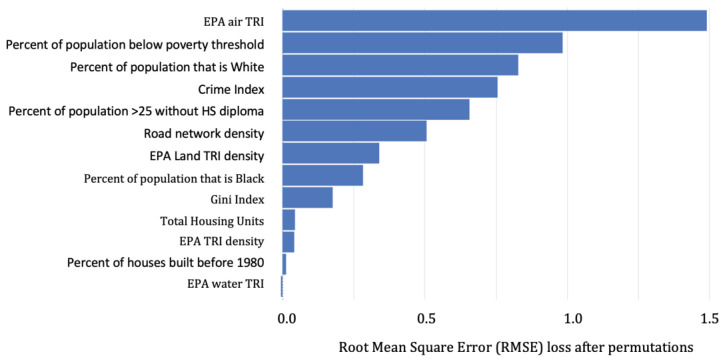
Predictor importance for ensemble learners predicting the number of children in a raster cell with BLLs ≥2–<5 µg/dL.

**Figure 3 ijerph-20-04477-f003:**
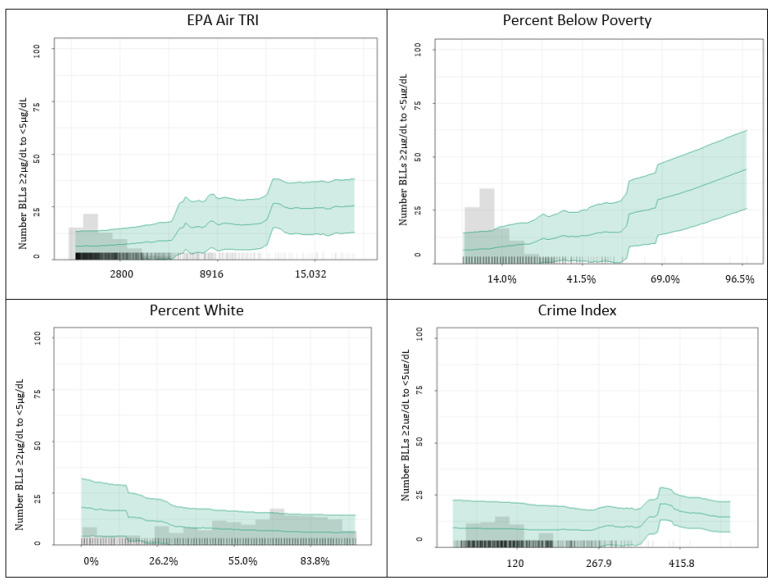
Partial dependence plots for top four ensemble predictors of the number of children in a raster cell with BLLs ≥2–<5 µg/dL.

**Figure 4 ijerph-20-04477-f004:**
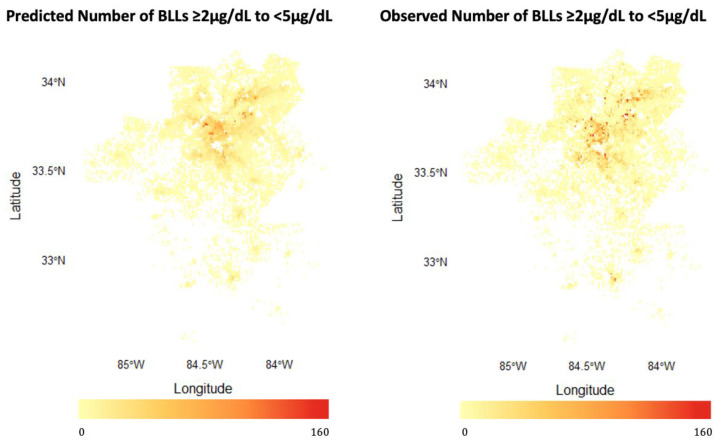
Raster grid of predicted and observed number of BLLs≥2–<5 µg/dL in the Atlanta metro area.

**Figure 5 ijerph-20-04477-f005:**
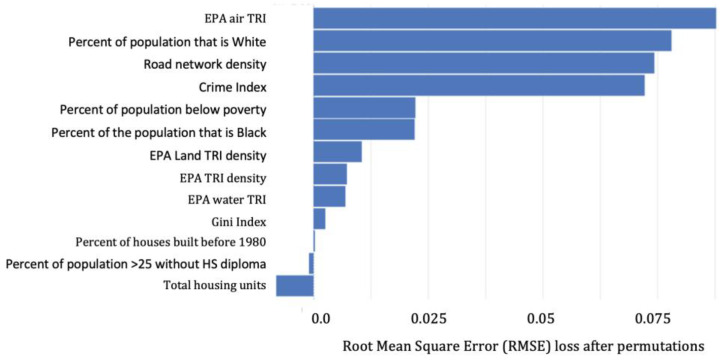
Predictor importance for ensemble learners predicting the number of children in a raster cell with BLLs ≥ 5 µg/dL.

**Figure 6 ijerph-20-04477-f006:**
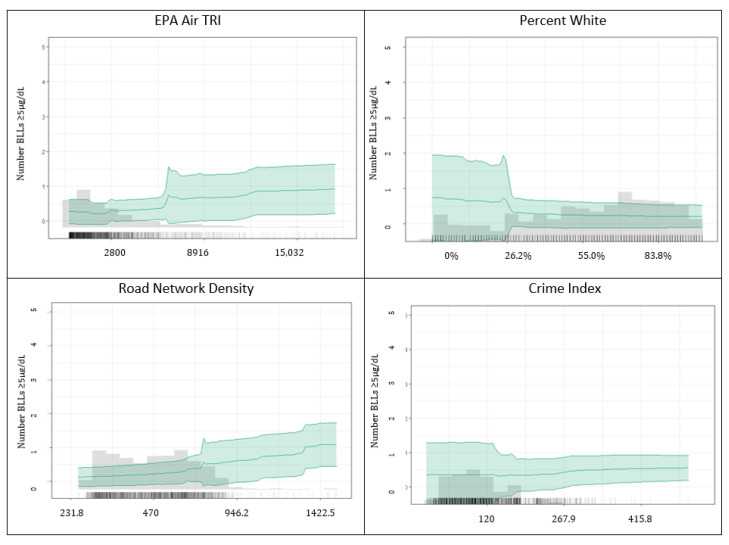
Partial dependence plots for top four ensemble predictors of the number of children in a raster cell with BLLs ≥ 5 µg/dL.

**Figure 7 ijerph-20-04477-f007:**
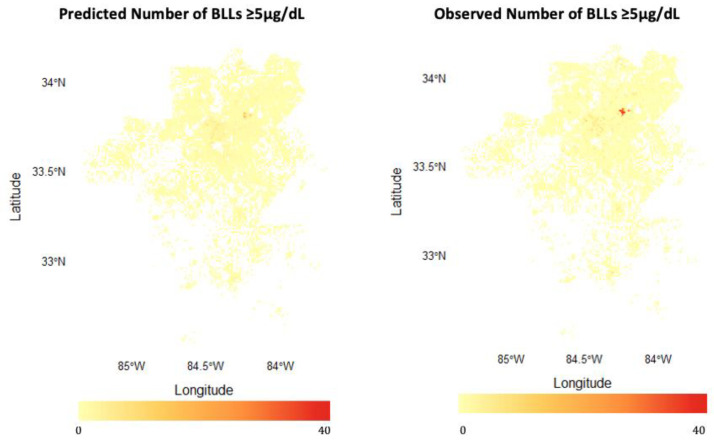
Raster grid of predicted and observed number of BLLs ≥ 5 µg/dL in the Atlanta metro area.

**Table 1 ijerph-20-04477-t001:** Sample demographic characteristics of the 92,792 urban and suburban children 0–72 months of age with a valid venous blood lead test (2010–2018) in the Atlanta metro area.

	<2 µg/dL (N = 38,124)	≥2 to <5 µg/dL (N = 52,522)	≥5 µg/dL (N = 2146)	Overall (N = 92,792)
Child age in months				
Mean (SD)	24.5 (14.0)	26.7 (15.3)	29.4 (16.3)	25.9 (14.8)
Median [Min, Max]	24.0 [0, 72.0]	24.0 [0, 72.0]	25.0 [0, 72.0]	24.0 [0, 72.0]
Child Sex				
Female	18,600 (48.8%)	25,468 (48.5%)	1020 (47.5%)	45,088 (48.4%)
Missing	36 (0.1%)	69 (0.1%)	6 (0.3%)	545 (0.6%)
Child Race				
Black	18,985 (49.8%)	12,006 (22.9%)	604 (28.1%)	31,595 (33.9%)
White	5272 (13.8%)	3507 (6.7%)	189 (8.8%)	8968 (9.6%)
Other	6288 (16.5%)	3536 (6.7%)	403 (18.8%)	10,227 (11.0%)
Not Reported	7491 (19.6%)	32,122 (61.2%)	927 (43.2%)	40,540 (43.5%)
Missing	88 (0.2%)	1351 (2.6%)	23 (1.1%)	1896 (2.0%)
Receiving Medicaid				
Yes	27,526 (72.2%)	39,413 (75.0%)	1456 (67.8%)	68,395 (73.4%)
Urban Zip Code				
Yes	7944 (20.8%)	12,519 (23.8%)	387 (18.0%)	20,957 (22.5%)

Note: SD—standard deviation; µg/dL—micrograms per deciliter.

**Table 2 ijerph-20-04477-t002:** Descriptive statistics of predictors and outcomes: 6591 raster cells in the Atlanta metro area.

Predictors	Mean, Median (Min–Max)
Number of children with BLLs >= 5 µg/dL	0.29, 0.0 (0.0–36)
Number of children with BLLs >2 to <5 µg/dL	7.2, 2.0 (0.0–63)
Percentage of the population below the poverty threshold	0.14, 0.11 (0.0–1.0)
Percentage of the population that is Black	0.36, 0.27 (0.0–1.0)
Percentage of the population that is White	0.55, 0.59 (0–1.0)
Percentage of houses built before 1980	0.31, 0.24 (0–1.0)
Percentage of the population >25 years old without an HS diploma	0.12, 0.10 (0–0.74)
Gini index	0.41, 0.41 (0.30–0.66)
Crime index	120, 98 (5.0–710)
Road network density	470, 470 (31–1600)
EPA TRI density	840, 380 (0–7100)
EPA Water TRI density	46,000, 32,000 (0–400,000)
EPA Air TRI density	2800, 1700 (0–18,000)
EPA Land TRI density	8200, 3000 (0–100,000)
Total Housing Units	1100, 980 (94–4200)

Note: TRI—Toxic release inventory; µg/dL—micrograms per deciliter; HS—high school.

**Table 3 ijerph-20-04477-t003:** Model performance when predicting the number of BLLs ≥2–<5 µg/dL and ≥5 µg/dL across base models for training, validation, and test data.

	BLLs ≥2–<5 µg/dL		BLL ≥5 µg/dL	
	RMSE	MAE	RMSE	MAE
Gradient Boosting Machine (GBM)				
Training	8.55	2.92	0.38	0.19
Validation	13.04	5.48	1.32	0.42
Test	15.51	6.03	1.57	0.46
Elastic Net Generalized Linear Model (GLM)				
Training	15.81	7.41	1.21	0.43
Validation	13.79	7.20	1.48	0.45
Test	17.07	7.60	1.65	0.45
Deep Neural Network (DNN)				
Training	11.32	5.42	1.19	0.37
Validation	12.96	5.62	1.44	0.39
Test	16.00	6.05	1.66	0.40
Ensemble Learner [Final Model]				
Training	9.11	3.93	0.62	0.23
Validation	12.74	5.53	1.33	0.38
Test	15.51	6.08	1.59	0.41
Simple Median [Comparison]				
Training	19.63	7.32	1.31	0.29
Validation	17.70	6.83	1.58	0.31
Test	20.29	6.93	1.72	0.30

## Data Availability

Data is not available due to sensitive participant information. The code is available upon request.

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
