# Peer review of "Predicting Low-Level Childhood Lead Exposure in Metro Atlanta Using Ensemble Machine Learning of High-Resolution Raster Cells"

_ijerph, 2023, doi:10.3390/ijerph20054477_

Round 1

Reviewer 1 Report

The manuscript uses ensemble machine learning of high-resolution raster cells to predict blood lead levels in children based on risk factors.  The novelty of this work is the ability of the model to predict blood lead levels on a sub-zipcode scale which allows the allocation of resources to the areas at most risk.  The model currently underpredicts the occurrence of blood lead exceedances.  However, the model shows potential and is a promising initial attempt at high-resolution blood lead prediction.  The manuscript is well-written and does not need significant revision.

The manuscript lists the strengths and limitations of the model.  One important limitation that was not emphasized is the percentage of participants that did not report race.  Race was not reported in 61% of those with BLL >2 to <5 ug/dL and 43% of those with a BLL >5 ug/dL.  The percent of the population that is white is within the top three predictors for both categories.  The reporting population may not accurately reflect the racial distribution in the area.  Because it has a large predictive effect on the model, the low response rate may be altering the results.  This should also be listed as a limitation and a possible reason why the model underpredicts actual BLL exceedances.

Author Response

Reviewer 1:

The manuscript uses ensemble machine learning of high-resolution raster cells to predict blood lead levels in children based on risk factors.  The novelty of this work is the ability of the model to predict blood lead levels on a sub-zipcode scale which allows the allocation of resources to the areas at most risk.  The model currently underpredicts the occurrence of blood lead exceedances.  However, the model shows potential and is a promising initial attempt at high-resolution blood lead prediction.  The manuscript is well-written and does not need significant revision.

Thank you for your kind review!

The manuscript lists the strengths and limitations of the model.  One important limitation that was not emphasized is the percentage of participants that did not report race.  Race was not reported in 61% of those with BLL >2 to <5 ug/dL and 43% of those with a BLL >5 ug/dL.  The percent of the population that is white is within the top three predictors for both categories.  The reporting population may not accurately reflect the racial distribution in the area.  Because it has a large predictive effect on the model, the low response rate may be altering the results.  This should also be listed as a limitation and a possible reason why the model underpredicts actual BLL exceedances.

This is an excellent point. We have added the following to the limitations section: “We should also note that a large percentage of children did not report race. Therefore, it is difficult to determine if our sample is representative by race in Atlanta. However, our model represents an important first step for high-resolution lead prediction using a very large sample with specific address information.” Please note that the individual-level race statistics were provided to explain the source of our data and provide information on generalizability of our model. Individual-level statistics were not included in our machine learning model.

Reviewer 2 Report

This is a nicely written, reasonably well described model of low level childhood lead exposure in Metro Atlanta. However it largely is a descriptive study and would benefit from a bit more compare and contrast with other more simplistic modeling approaches, such as inverse distance weighting, simple median values, and perhaps even kriging, and kriging with a trend. What we really want to know is not just descriptions of this model, but whether or not this model performs better than other models.  Correlation coefficients can be used comparing predicted with observed values; or weighted kappa statistics/sensitivity/specificity/PPV/NPV to compare categories.

The methods could be a little better described. 2.1: Were these locations geocoded at current residence? 2.3.1: how was the TRI score generated; more details are need. 2.3.1 (the second one) should likely be 2.3.2, crime index. For this section on crime, do any raster cells fall within multiple census blocks? If so, do you take a weighted average from the census blocks? 2.4.2: some basic description of the different algorithms would be useful.

Top of p. 6: typo: a clause comes before the start of the sentence. Needs editing.

Results: It seems the GBM approach yields the lowest RMSE. Maybe it’s a better approach than the ensemble learner?  This should be discussed, ideally within the context of comparing observed vs predicted models as requested above.

Good point about limitations in the data that may hinder generalizability.

Author Response

Reviewer 2:

This is a nicely written, reasonably well described model of low level childhood lead exposure in Metro Atlanta. However it largely is a descriptive study and would benefit from a bit more compare and contrast with other more simplistic modeling approaches, such as inverse distance weighting, simple median values, and perhaps even kriging, and kriging with a trend. What we really want to know is not just descriptions of this model, but whether or not this model performs better than other models.  Correlation coefficients can be used comparing predicted with observed values; or weighted kappa statistics/sensitivity/specificity/PPV/NPV to compare categories.

Thank you for your kind review! We agree that additional contrasting modeling approaches would be useful. We decided to include the simple median of the data as a comparative measure for RMSE and MAE. Kriging is an interesting option. However, because kriging uses existing knowledge about the locations of points of each participant we cannot compare this model with our model. Our model does not have pre-existing knowledge of the locations of participants. It is only using variables gleaned from the geographic area to predict both location and number of children in a raster cell with lead exposure. We have clarified this point in the discussion.

You will see in Table 3 that the ensemble predicts better than the median via RMSE but not necessarily much better using MAE for BLL >5ug/dL. This makes sense because we are using very skewed count data. RMSE gives an extra penalty if the observation is much larger than the predicted value. This is a comparison that is important for our prediction. MAE doesn’t penalize the observed-predicted difference as strongly. We have added the median comparison in the results section and this clarification about RMSE in the discussion section.

The methods could be a little better described. 2.1: Were these locations geocoded at current residence? 2.3.1: how was the TRI score generated; more details are need. 2.3.1 (the second one) should likely be 2.3.2, crime index. For this section on crime, do any raster cells fall within multiple census blocks? If so, do you take a weighted average from the census blocks? 2.4.2: some basic description of the different algorithms would be useful.

Thank you for these corrections. We have followed your suggestions and added more clarity in the noted sections, especially to variable creation and the ensemble algorithms. For all raster cells, we used the centroid of the raster to assign it to the census tract. While some cells may have fallen within multiple census tracts the centroid approach assigns the census tract where the majority of the cell is within.

Top of p. 6: typo: a clause comes before the start of the sentence. Needs editing.

Thank you. This has been removed.

Results: It seems the GBM approach yields the lowest RMSE. Maybe it’s a better approach than the ensemble learner?  This should be discussed, ideally within the context of comparing observed vs predicted models as requested above.

While it’s true that the GBM is performing closely to the ensemble, that is to be expected. The GBM model informs the ensemble and the other models help prevent overfitting. This improves model performance when predicting with new sets of data. The overall purpose of Table 3 is to demonstrate that the ensemble is performing well and that there isn’t significant overfitting (differences in predictive performance between training, validation and test datasets). This clarification has been added to the discussion.

Good point about limitations in the data that may hinder generalizability.

Thank you!

Round 2

Reviewer 2 Report

Reviewers' comments were addressed.